

# What could irrigated agriculture mean for Amazonia? A review of green and blue water resources and their trade-offs for future agricultural production in the Amazon Basin

Michael J. Lathuillière[1], Michael T. Coe[2], Mark S. Johnson[1,3]

[1]Institute for Resources, Environment and Sustainability, University of British Columbia, 2202, Main Mall, Vancouver, B.C., V6T 1Z4, Canada
[2]Woods Hole Research Center, 149 Woods Hole Road, Falmouth, MA, 02540-1644, U.S.A
[3]Department of Earth, Ocean and Atmospheric Sciences, University of British Columbia, 2207, Main Mall, Vancouver, B.C., V6T 1Z4, Canada

*Correspondence to*: Michael J. Lathuillière (mlathuilliere@alumni.ubc.ca)

**Abstract.** The Amazon Basin is a region of global importance for the carbon and hydrological cycles, a biodiversity hotspot, and a potential centre for future economic development. The region is also a major source of water vapour recycled into continental precipitation through evapotranspiration processes. This review applies an ecohydrological approach to Amazonia's water cycle by looking at contributions of water resources in the context of future agricultural production. At present, agriculture in the region is primarily rain-fed and relies almost exclusively on green water resources (soil moisture regenerated by precipitation). Future agricultural development, however, will likely follow pathways that include irrigation from blue water sources (surface and groundwater) as insurance from variability in precipitation. In this review, we first provide an updated summary of the ecohydrological framework before describing past trends in Amazonia's green and blue water resources within the context of land use and land cover change. We then describe green and blue water trade-offs in light of future agricultural production and potential irrigation to assess costs and benefits to terrestrial ecosystems, particularly land and biodiversity protection, and regional precipitation recycling. Management of green water is needed, particularly at the agricultural frontier located in the headwaters of major tributaries to the Amazon River, and home to key downstream blue water users and ecosystem services, including domestic and industrial users, as well as aquatic ecosystems.

## 1 Introduction

The role of ecosystems in the global hydrological cycle has been the foundation of global ecohydrology over the past 50 years (Dolman et al., 2014). Advances in the areas of remote sensing and land-atmosphere modelling have widened our understanding of ecosystems in the global carbon and hydrological cycles, and identified important global trends in evapotranspiration (ET). These trends include an apparent slowdown in global ET in recent decades and possible increase in continental ET in South America (see Supplemental Information). The global ET decline has been attributed to increased



atmospheric $CO_2$ concentrations and nitrogen deposition, but also changes in land use and soil moisture stocks (Jung et al., 2010; Mao et al., 2015).

Soil moisture plays an important ecohydrological role. To highlight its importance, Falkenmark and Rockström (2004) proposed to shift the traditional notion of the freshwater source from surface or groundwater resources to precipitation. As

precipitation reaches the soil surface, it is partitioned into two distinct resources: "blue" water represents runoff, rivers, reservoirs, wetlands and aquifers (liquid stocks); "green" water is the soil moisture found in the soil's unsaturated zone either reclaimed by the atmosphere through soil evaporation, or consumed by the vegetation via root uptake and lost through transpiration during photosynthesis (Falkenmark and Rockström, 2004). Precipitation over land is recycled exclusively through ET processes, of which about two thirds are supplied by plant transpiration (Gerten et al., 2005; Rost et al., 2008), thus making

green water an essential ecohydrological resource that merits scientific investigation.

We propose to evaluate this framework as a means for analysing water resources in the Amazon region in the context of food production and security, freshwater availability and global climate change. In recent decades, the region has experienced significant deforestation for agricultural expansion of soybean, maize and pasture (Fearnside, 2005; Barona et al., 2010; Macedo et al, 2012; Nepstad et al., 2014) while exhibiting the effects of apparent changes in climate from El Niño events, two

historical droughts in 2005 and 2010 (Davidson et al., 2012) and possible land use driven atmospheric feedbacks affecting precipitation (Spracklen et al., 2012; Bagley et al., 2014; Spracklen and Garcia-Carrera, 2015). South-eastern Amazonia, in particular, is sensitive to future anthropogenic and climate changes (Coe et al., 2013). Its location in the headwaters of major tributaries of the Amazon River and a key node in the global food system also make it an important geographical player in land and water management.

Southern Amazonia is a regional hotspot for potential trade-offs in green and blue water resources between upstream and downstream users. Agricultural expansion and especially the use of irrigation remain important options for consideration this century as these trade-offs are intrinsically linked to land management decisions. South-eastern Amazonia currently has high crop yields, comparable to those in the United States, of about 3 tonnes ha$^{-1}$ for soybean and 5–6 tonnes ha$^{-1}$ for maize (IBGE, 2015). Therefore, production can increase only marginally through improvements in rain-fed practices. Deforestation and

agricultural expansion in Amazonia have been thoroughly examined using land use change trajectories and including the potential effects of atmospheric $CO_2$ concentration on future agricultural yields and river discharge (e.g. Coe et al., 2009' Coe et al., 2011; Oliveira et al., 2013; Pokhrel et al., 2014), but these studies have not examined possible expansion of the use of irrigation in agriculture and the impacts that this freshwater consumption could have on groundwater resources, river discharge and climate.

We review what the green and blue water ecohydrological approach can bring to land and water management in the context of agricultural production before describing implications of green/blue water trade-offs with land use change in Amazonia, particularly South-eastern Amazonia. We discuss possible regional changes to the water cycle and atmospheric water balance as a result of conversion of natural ecosystems and pasture to agricultural production and the adoption of widespread irrigation.





The review concludes by providing an assessment highlighting green and blue water trade-offs in light of possible agricultural production pathways as a way to stimulate discussion and inspire future research.

## 2 Green and blue water as a foundation for ecohydrology

### 2.2 Defining an ecohydrological paradigm for water resources

Falkenmark and Rockström (2004) initiated a paradigm shift in water resource management by proposing to change the traditional notion of the freshwater source from rivers, reservoirs and aquifers to precipitation. In their description of water resources, precipitation is partitioned at the soil surface into blue water (as surface or groundwater), and green water (as soil moisture regenerated by precipitation) typically absent in water management considerations (Falkenmark and Rockström, 2004, 2006). While blue water has been subject to millennia of human management and engineering history, the idea of green

water management is fairly recent (Falkenmark and Rockström, 2006). Traditional blue water management considers ET as a flow of water lost to the atmosphere (Oki and Kanae, 2006), while green water resource management calls for a focus on vapour supply to the atmosphere for precipitation recycling (Falkenmark and Rockström, 2004, 2006; Ellison et al., 2012). On the land, such vapour supply is represented by green water consumed mainly by ET, but also some unconsumed water that is returned to the atmosphere through evaporation from soil moisture, snow or ice sublimation, and evaporation of water

intercepted by human made or natural landscapes.

Blue and green water are distinguished by their physical state as well as the processes, frequency, and factors of influence that govern their consumptive uses (Table 1). Consumptive uses are different from water withdrawals in that withdrawals can be returned to the blue water cycle, whereas consumptive uses cannot (Rockström et al., 2010). Blue water consumptive uses include some fraction of drinking water, evaporative losses through cropland irrigation or hydropower, as well as incorporation

of water into products. Green water consumptive uses exclusively occur through ET with a distinction between productive and unproductive vapour flows characterized respectively by transpiration and direct evaporation of soil moisture (Falkenmark and Rockström, 2006).

Aquatic ecosystems rely exclusively on blue water and may require inflows of surface or groundwater to ensure proper function (e.g. wetlands, fisheries). Blue water is shared between humans and ecosystems such that a consumption activity will require

blue water trade-offs between users. Green water resources are exclusively consumed through ET processes and, as such, are consumed only once by terrestrial ecosystems, in the case of productive green water consumption, before returning to the atmosphere (Rockström and Gordon, 2001). Evaporation of soil moisture or water intercepted by canopies also regenerate precipitation unproductively, meaning the process does not support any additional human or ecosystem activity, although the latent heat uptake from evaporation can be important to regional water balances (Biggs et al., 2008). Water vapour flows

through a blue water redirect (Karlberg et al., 2009) illustrate the special case of ET resulting from irrigation, a blue water resource. In order to separate the resource "colours", this special case is qualified as a blue water consumptive use (Rockström et al., 2010).



The above representation of the water source as precipitation and its partitioning into blue and green water resources bring new considerations to water resource management: the importance of vapour flows and precipitation recycling, as well as potential blue and green water trade-offs with land management discussed below.

## 2.2 Green water resources, vapour flows and precipitation recycling

By emphasizing the importance of green water resources, Falkenmark and Rockström (2004) highlighted the need to regenerate rainfall through ET to ensure a continuation in the supply of precipitation, the water source in the above paradigm. Precipitation recycling occurs at different scales when considering local (e.g. same watershed), regional or continental effects since regional terrestrial recycling is generally larger than local recycling (40% and 13% respectively (Ellison et al., 2012), and increases as a function of area (van der Ent et al., 2010). It is estimated that about 65% of global terrestrial precipitation is sourced from terrestrial water vapour flows to the atmosphere (Karlberg et al., 2009) (Table S1), with the remaining 35% sourced by evaporation from oceans and other water surfaces (liquid, ice and snow) (Oki and Kanae, 2006). Green water resources therefore recharge "atmospheric watersheds" or "precipitationsheds" which connect soil moisture evapotranspired from one source region to a sink region, further downwind (Keys et al., 2012). As ET is typically water-limited in arid, semi-arid, and highly seasonal environments, these regions are particularly reliant on green water resources to regenerate precipitation (Falkenmark and Rockström, 2004, 2006; Rockström et al., 2010).

Key terrestrial ecosystem services are maintained through the consumption of green water resources, including agro-ecosystems, with vapour flows ensuring 90% of global human needs (Rockström and Gordon, 2001). Global terrestrial ecosystems were estimated to supply 42,900–45,646 $km^3$ $y^{-1}$ of water vapour through transpiration, while 14,682–15,478 $km^3$ $y^{-1}$ of water vapour is predicted to be supplied by rain-fed agroecosystems (cropland and pasture) (Gerten et al., 2005; Rost et al., 2008). Therefore, there is an intimate link between land and water resources planning (Falkenmark and Rockström, 2006) with green water acting as a major source of vapour supply, and an essential resource in tackling global ecohydrological challenges such as food security (Rockström et al., 2010), carbon sequestration, drought mitigation and climate change resilience (Dolman et al., 2014).

## 2.3 Precipitation partitioning on land and green/blue water trade-offs

Since green and blue water resources emerge through the partitioning of precipitation at the soil interface, there are green/blue water trade-offs with every land use decision (Karlberg et al., 2009). Several models have attempted to provide annual average ET estimates for major biomes of the world (Table 2) which highlight partitioning of precipitation and potential trade-offs based on vegetation and climate. These trade-offs depend on the ecohydrological relationship between the vegetation and the water cycle which can be explained through environmental and physiological controls on ET. Environmental controls are illustrated by the relationship linking energy, climate, vegetation, and land use. In 299 basins, Zeng et al. (2012) found strong correlations between ET and mean annual temperature (R = 0.68), annual precipitation (R = 0.87) and NDVI (R = 0.70). Analysis of 21 tropical eddy covariance sites showed strong correlation of latent heat flux with net radiation ($R^2 = 0.72$), vapour



pressure deficit ($R^2 = 0.14$), and NDVI ($R^2 = 0.09$) (Fisher et al., 2009). These relationships show the dependence of ET on green water supply and environmental demand (*sensu* Christoffersen et al., 2014), and identifying exactly which mechanism guides ET in a given landscape is the essential ecohydrological question (Jung et al., 2010).

Changing the landscape can affect both environmental and physiological controls on ET with consequences on green/blue water trade-offs. Shallower root systems of crops or pasture compared to woodlands or tropical forest is one important morphological condition that explains differences in precipitation partitioning (Table 2) in addition to the amount of water supplied by precipitation or access to deep groundwater (blue water) reserves (Matyas and Sun, 2014). Declines in green water from the landscape due to changes in climate or land use can lead to an increase in blue water downstream due to runoff (Karlberg et al., 2009) and consequently a reduction in moisture recycling to regenerate precipitation locally (Savenije, 1995; 1996). However, the scale of these processes is an important consideration since reduced precipitation recycling can also affect blue water by lowering rainfall inputs into rivers (Ellison et al., 2012).

Research linking global agricultural practices to changes in vapour flows have emphasized the importance of deforestation and irrigation expansions in some regions with expected consequences on precipitationsheds (Keys et al., 2012). Deforestation was estimated to have reduced transpiration by over 100 mm y$^{-1}$ in regions of intensive land use change with a global average of 3032 mm y$^{-1}$ of reduced transpiration modelled for the 1961–1990 period (a 7.4% and 9.4% decrease in transpiration and evaporation respectively) (Gerten et al., 2005). Another study concludes that agricultural expansion through deforestation led to a global water vapour loss of 3000 km$^3$ y$^{-1}$, while irrigation expansion has increased flows by 2600 km$^3$ y$^{-1}$ suggesting a global compensation of ET with land cover and uses (Gordon et al., 2005). In parallel, Gerten et al. (2005) also predict a 2.2% increase in runoff, while Rost et al. (2008) estimate a 5% increase in river discharge accompanying a 2.8% decrease in ET from land use change and a 1.9% increase in ET from irrigation. Such trade-offs would tip the balance toward greater blue water yields. However, the question of scale needs to be addressed as precipitation recycling might counteract this effect and so, just like global changes in ET, global changes in runoff might not represent regional effects. Reductions in ET can lessen precipitation hundreds or thousands of kilometres away, thus also impacting river discharge (Coe et al., 2009, 2011; Ellison et al., 2012; Spracklen et al., 2012; Stickler et al., 2013; Spracklen and Garcia-Carrera, 2015).

## 2.4 The role of green water in agriculture

As the largest consumer of water resources, agriculture has been the focal point of early research on green and blue water. With 805 million people still chronically undernourished (FAO, 2014), the question often asked is: how can water resources be managed efficiently to feed the world given resource constraints while considering changing diets, fish production, animal feed and biofuel projections? (Gordon et al., 2005 ; Hoff et al., 2010 ; de Fraiture, 2007 ; Rockström et al., 2007, 2009, 2010, 2014). An additional 1700 km$^3$ y$^{-1}$ and 1550 km$^3$ y$^{-1}$ of water consumptive use is expected for increases in food production and carbon sequestration projected for 2050, respectively, compared to the current total blue water consumptive use of 2600 km$^3$ y$^{-1}$ (Rockström et al., 2014). This combined 5800 km$^3$ y$^{-1}$ approaches the upper limit of the estimated planetary boundary of



4000–6000 km$^3$ y$^{-1}$ (Rockström et al., 2014), and given that blue water resources are already stressed in many regions of the world, there seems to be limited opportunity to feed the world solely through irrigation expansion.

Green water consumption in rain-fed agriculture represents about 75% of total cropland consumptive use (green and blue water), which is four to five times greater than blue water consumptive use in irrigation according to seven global models

(Hoff et al., 2010). Estimates of global cropland ET were predicted to be between 3272 km$^3$ y$^{-1}$ and 7200 km$^3$ y$^{-1}$ based on models considered while pasture ET exceeds 4000 km$^3$ y$^{-1}$. Rain-fed agriculture is often key to securing the livelihoods of those living in poverty, especially in drylands or savannah regions where crop water requirements typically exceed precipitation (Rockström et al., 2009). Such regions are not necessarily considered water scarce, rather it is the intensity and timing of precipitation throughout the year and its concentration in short wet seasons, which presents challenges for land and

water resources management. As such, these regions, especially sub-Saharan Africa, have been the focus of research on upgrading rain-fed agriculture: under improved management, the current 10–30% use of green water could increase to 50% with significant increases in yields (Falkenmark and Rockström, 2006). Such a strategy aims to reduce evaporation (unproductive green water) and increase transpiration (productive green water) through a so called "vapour shift" (Rockström, 2003; Rockström et al., 2007).

Improvements in water productivity, or the amount of crops produced per unit input of water consumed (Cai et al., 2011), could decrease the water requirements for food production in 2050 by almost 2850 km$^3$ y$^{-1}$ broken down into 725 km$^3$ y$^{-1}$ of blue water (i.e. irrigation) and 2125 km$^3$ y$^{-1}$ secured through green water resources (i.e. rain-fed agriculture) (Rockström et al., 2010). Strategies for increasing food production and reaching self-sufficiency differ based on the green and blue water resource potential of each country (Rockström et al., 2009). An increase in the use of blue water resources for irrigation remains a viable

option as long as it does not promote further water scarcity, and does not impose damages to aquatic ecosystems or land subsidence. Some countries have the potential to increase their yields by improving rain-fed agriculture (e.g. Kenya, Uganda, and Ethiopia) (Rockström et al., 2009). The expansion of rain-fed agriculture into native terrestrial ecosystems remains another option for countries that could significantly expand green water resources for food production, although not without affecting biodiversity and precipitation recycling while increasing local runoff. Virtual water imports, or the import of water virtually

via agricultural products from national and international trade, is the final option for countries that are already chronically blue and green water short (e.g. Jordan, Israel, Pakistan, Iraq) (Rockström et al., 2009), and a strategy which has been under scrutiny in order to qualify water savings from trade (Dalin et al., 2012; Hoekstra and Mekonnen, 2012).

South-eastern Amazonia (Brazil) is strongly seasonal, with semi-arid conditions during extended periods leading to very different green and blue water realities than those described above. Brazil's economic future is focused on the continuous

increase of production for export of soybean and maize feed for cattle, and beef (MAPA, 2013). This frontier is located in the headwaters of main tributaries of the Amazon River, in which fisheries, navigation and hydroelectric projects are important downstream blue water users. For example, the 176,000 km$^2$ Upper Xingu River Basin of Mato Grosso contains over 22,000 springs feeding the 510,000 km$^2$ Xingu Basin (Figure 1) (Velasquez and Bernasconi, 2010; Macedo et al., 2013) and may be home to the future Belo Monte dam which will require significant amounts of blue water to operate (Stickler et al., 2013). As



a major production centre for commodities, increases in agricultural production will need to consider green and blue water trade-offs from possible production pathways such as expansion into natural ecosystems, expansion into pastureland, or intensification into current land, along with additional irrigation as insurance for dry spells and drought years. The additional water vapour supply from irrigation as well as other upstream water bodies (e.g. small farm dams) represents an important

planning consideration for the regional water cycle. While much of previous research has focused on regional temperatures and greenhouse gases (Oliveira et al., 2013), precipitation recycling (Stickler et al., 2013; Bagley et al., 2014), river discharge (Coe et al., 2011, 2009; Panday et al., 2015), and impacts to biodiversity (Chaplin-Kramer et al., 2015), detailed modelling studies on how potential increases in regional water vapour flows from irrigation may impact the water cycle in Amazonia are still lacking despite the current state of knowledge on atmospheric feedbacks from land use change. We explore these

implications by focusing exclusively on green/blue water trade-offs in the region.

## 3 Land use change as a driver of green/blue water trade-offs in Amazonia

### 3.1 Controls on evapotranspiration in Amazonia

The Amazon Basin is abundant in both green and blue water (see Supplemental Information) whose trade-offs result from environmental and biological controls of ET. Environmental controls follow a precipitation gradient that declines from North

to South over the 0–11°S latitudinal band (Manaus to Sinop in Table S3, Figure 1). In equatorial Amazonia (e.g. Manaus, Santarem), ET seasonality is primarily driven by radiation, but also morning fog especially in the wet season (Anber et al., 2015). The dry season occurs later in the calendar year (July–November) when increasing solar radiation coincides with limited cloud cover favouring photosynthesis and increasing ET to more than 100 mm month$^{-1}$ (Restrepo-Coupe et al., 2013; Christoffersen et al., 2014). In equatorial Amazonia, latent heat flux is highly correlated to net radiation ($R^2 = 0.53$) suggesting

that available energy is a strong control on ET within the latitudinal band (Restrepo-Coupe et al., 2013). With little or no soil moisture stress affecting the productivity of broadleaf evergreen forests, ET in equatorial Amazonia is only mildly seasonal as green water stocks remain largely available for ecosystems to consume all year round.

In contrast, ET in Southern Amazonia is strongly seasonal. Remote sensing observations from MOD16 (2000–2009) for the Amazon-Cerrado transition forest showed a forest-wide ET of 65 mm month$^{-1}$ for August periods, only 60% of rainy season

ET values of 105–115 mm month$^{-1}$ between January and April (Lathuillière et al., 2012). Future increases in regional temperatures could lead to an overall basin wide increase in ET due to an increase in potential ET, limited, however, by regional differences in soil moisture availability, as well as groundwater reserves which can be deeper than 20 meters in South-eastern Amazonia (Pokhrel et al., 2014).

Biological controls on ET have been shown to occur in Southern Amazonia's transition forest where high vapour pressure

deficit in the dry season can trigger stomatal closure and allow forest ecosystems to conserve water in water limited conditions (Costa et al., 2010). Access to green water by deeply rooted trees has been suggested as a drought resilience mechanism for forest ecosystems in the region with roots accessing soil moisture over eight meters deep (Nepstad et al., 1994; Davidson et



al., 2011). Deeply rooted trees help sustain ET over Southern Amazonia's dry season (Coe et al., 2009, 2011; Lathuillière et al., 2012; Christoffersen et al., 2014; Biudes et al., 2015, Panday et al., 2015; Silvério et al., 2015; Vourlitis et al., 2015) and will likely become more important with increased air temperatures (Pokhrel et al., 2014). As such, land use change resulting in the replacement of forest by more shallow rooted pasture grasses or cropland reduces the amount of accessible green water

and vapour flows to the atmosphere.

Given these well-defined processes across the basin and the important role of seasonality in the Southern portion, South-eastern Amazonia appears as a region that requires special attention. The region's ET processes are water limited during an extended dry season. A rise in the local dry season temperatures show the importance of soil moisture and groundwater as important water sources for deeply rooted trees to ensure continuous water vapour flows to the atmosphere (Pokhrel et al., 2014). Its

geographical importance, both as the home to Brazil's expanding agriculture and the region upstream of the Amazon River, make it an environmentally and economically important region that is sensitive to future land use and climate changes. Green and blue water trade-offs will be inevitable considering the current and future land use changes which decrease green water consumption of terrestrial ecosystems and can increase blue water through runoff.

## 3.2 Land use change activity for agricultural production

Brazil's internal colonization driven by the Agrarian reform of the 1960s, brought intensive agricultural activity to the Amazon and Cerrado regions. The 1980s and 1990s saw the expansion of settlements in the region, starting with cattle ranching later followed by soybean production, both of which created economic activity that also required road building and ever increasing mechanization of deforestation and agriculture (Fearnside, 2001, 2005; Morton et al., 2006; Rudel et al., 2009). The country's expansion of soybean production followed a South to North progression into the Cerrado and closer to the Amazon biome

(Simon and Garagorry, 2005) today reaching South-eastern Amazonia and the state of Mato Grosso (Figure 1). Soybean expansion has occurred either directly through a forest to cropland conversion, or indirectly through a pasture transition (Macedo et al., 2012; Spera et al., 2014; Silvério et al., 2015), which has displaced pasture further North into the Amazon (Barona et al., 2010).

The state of Mato Grosso (Figure 1) is a hotspot for this expanding agricultural frontier with more than a decade of documented

deforestation activity (Macedo et al., 2012). In accordance with land use change practices, government policies and private initiatives, Nepstad et al. (2014) identified three distinct phases guiding deforestation: Agro-industrial expansion (pre-2004), Frontier Governance (2005–2008) and Territorial Performance (2009–present). Mato Grosso and Pará have shown the greatest rates of deforestation in Amazonia with an accumulated 138,289 km$^2$ and 137,923 km$^2$ respectively for the 1988–2014 and, together, contributed to 70% of total deforestation in Brazil (INPE, 2015). The Brazilian Federal Forest Code is the main

Federal legislation controlling deforestation in Brazil. The 1965 version of the law requires a land reserve in which 80% of native vegetation must be retained on properties located in the Amazon biome, but this requirement on native vegetation retained drops to 50% for the Amazon-Cerrado transition zone and 20% for the Cerrado (Brannstrom et al., 2008; Soares-Filho et al., 2014). A new version of the Forest Code was signed in 2012, which retains the old reserve, provides new rules for illegal





deforestation prior to 2008 while adding new incentives to reduce deforestation such as trade in land reserves between properties (Soares-Filho et al., 2014).

The drop in Mato Grosso's deforestation rates in the late 2000s coincided with a drop in exchange rate of the Brazilian Real which increased the opportunity costs of deforestation (Richards et al., 2012), restrictive access to credit for producers located in municipalities labelled as hotspots of deforestation, as well as a Soybean Moratorium (2006) and a Cattle Agreement (2009) that sought to remove any suppliers from the soybean and meat supply chains that have produced on land previously cleared from forests (Macedo et al., 2012; Nepstad et al., 2014; Gibbs et al., 2015). Soybean and beef production, however, continued to grow with further internationalization of commodity markets as China imported an ever increasing amount of soybean to meet its increasing national demand, mostly for producing animal protein (Lathuillière et al., 2014). Deforestation has been apparently on the rise since 2012 (INPE, 2015) which coincides with the implementation of the new Forest Code (Soares-Filho et al., 2014; Spracklen and Garcia-Carrera, 2015). Land use change activities have been recognized to have an effect on the local climate (Davidson et al., 2012) with emerging evidence of changes in regional and continental precipitation recycling with South-eastern Amazonia playing an important role in the Amazon Basin (Table S4 in the Supplemental Information).

## 3.3 Land use change effects on the water balance

Differences in the energy balance have been observed on different landscapes across Amazonia (Table S3). Therefore, land use change from one landscape to another is expected to affect radiation partitioning with noted impacts on the water cycle. Model simulations in South-eastern Amazonia have shown that changes in land cover affect surface albedo, while morphological (vegetation height, root depth, albedo) and physiological changes (C3 to C4 photosynthetic pathways) can affect the magnitude of sensible and latent heat fluxes (positively and negatively) with possible effects on surface temperature (Pongratz et al., 2006; Davidson et al., 2012; Bagley et al., 2014). Analysis of satellite information obtained for the Upper Xingu River Basin of Mato Grosso showed that forest to cropland and forest to pasture land use transitions in the 2000s decreased ET (32% and 24%), increased sensible heat flux (6% and 9%) and increased surface temperature up to 6.4°C (Silvério et al., 2015). In the Amazon Basin, deforestation reduced ET by 5%, increased sensible heat flux by 2% and decreased precipitation by 6% in the dry season (Bagley et al., 2014), all of which were exacerbated in drought years (6%, 4% and 6% respectively). Morphological differences in the root infrastructure can make green water more accessible to maintain ET processes during the dry season (Nepstad et al., 1994; Lathuillière et al., 2012).

The above changes in surface energy balance affect the partitioning of precipitation into blue and green water as quantified by runoff and ET. Field studies in South-eastern Amazonia have shown that soybean watersheds can have water yields up to four times greater than forested watersheds (Hayhoe et al., 2011; Dias et al., 2015). Coe et al. (2009) simulated runoff in the Amazon Basin through a coupled land-atmosphere and climate change numerical model. Most tested river basins exhibited an increase in discharge for 2000 and 2050 when compared to potential natural vegetation, even in a restrictive deforestation governance scenario. The Tocantins and Madeira rivers (Figure 1) saw discharges increase from 26% and 7% in 2000 to 18% and 32% for





2050 respectively (Coe et al., 2009). Similarly, discharges of the Xingu and Araguaia rivers have increased 6% (1970 to 2000s) and 25% (1970 to 1990) primarily due to deforestation and climate (Coe et al., 2011; Panday et al., 2015).

Local changes in land cover also change vapour supply to the atmosphere, which can reduce regional precipitation and, indirectly, river discharge in the basin (Coe et al., 2009, 2011; Ellison et al., 2012). The Amazon Basin is the source of moisture to a precipitationshed that provides subtropical rainfall as far south as the La Plata Basin through the South American Low Level Jet (Marengo, 2006; Keys et al., 2012). Vegetative surfaces promote additional vapour inputs into air masses that result in precipitation in downwind areas (Spracklen et al., 2012) with evaporated water sources contributing to continental precipitation less than 2000 km away (van der Ent and Savenije, 2011). Results for the 2000–2009 period show a 10% drop in the contributions of forests to total water vapour flows to the atmosphere (a decrease of 119 km$^3$ from 593 km$^3$ y$^{-1}$ to 474 km$^3$ y$^{-1}$) due to a shift of green water use from natural ecosystems to agricultural production in the state of Mato Grosso (Lathuillière et al., 2012). In the same time period, the Upper Xingu Basin experienced a 35 km$^3$ y$^{-1}$ ET drop due to land use change (Silvério et al., 2015). This reduction in vapour supply to the atmosphere can also affect river discharges and hydropower generation within the basin (Stickler et al., 2013).

Local land cover change compounds the effects of inter-annual variability effects of regional precipitation in the basin. While pluvial and drought years affected regional precipitation regimes, local deforestation impacts on precipitation were at least as important as regional effects (Bagley et al., 2014). Areas of deforestation showed up to a 20% decreases in precipitation during the dry months of drought years (Bagley et al., 2014) with interconnected regions of precipitation source, or precipitationshed (e.g. Central and Southern Amazonia) to distant sinks (North-western Amazonia). On a local scale, precipitation in the Xingu River Basin was found to be sensitive to potential future deforestation both inside the confines of the basin and in the rest of the Amazon forest (Stickler et al., 2013). This means that land cover change in one region can greatly affect precipitation in addition to local recycling, such as in South and Central Amazonia (Spracklen et al., 2012; Bagley et al., 2014). Drought years were found to increase recycled evaporation from 67% to 74% in the dry months of South-eastern Amazonia (Bagley et al., 2014) indicating that atmospheric water demand can be met, in part, from regional sources.

### 3.4 Linking vapour supply to precipitation and terrestrial ecosystems

A diminished vapour supply to the atmosphere as a result of land use change can affect regional precipitation patterns which can in turn impact ecosystem processes and services in the region. Precipitation has been declining in Amazonia in recent years (Hilker et al., 2014). Analysis of 280 meteorological stations across the basin showed a decline in precipitation of $5.3 \pm 0.7$ mm y$^{-1}$ for the 1996–2005 period, and an increase to $7.8 \pm 1.6$ mm y$^{-1}$ in areas with denser tree cover (Brando et al., 2010). More recent analysis of satellite imagery confirmed a 17–30% decline in precipitation over the greater portion of landscapes of the Amazon region for the 2000–2012 period, especially in Eastern and South-eastern Amazonia which showed a 25% decline in precipitation for the period (Hilker et al., 2014).

In contrast, the Cerrado did not show any decline in precipitation between 2002 and 2010, but the biome did see an average increase in ET of $51 \pm 15$ mm y$^{-1}$ (Oliveira et al., 2014). The above results are in line with a review of 26 studies linking



deforestation to reductions in precipitation (Marengo, 2006) as well as 96 simulations (Spracklen and Garcia-Carrera, 2015). Deforestation is also known to increase the length of the dry season, particularly in Southern Amazonia, with possible changes in the onset of the wet season (Costa and Pires, 2010). Since 1979, the end of the dry season in Southern Amazonia has been delayed by $4.5 \pm 2.0$ days per decade with serious implications on the integrity of the tropical forest ecosystem should the dry season length continue to increase this Century (Fu et al., 2015).

Deforestation in a business-as-usual scenario could lead to declines in precipitation by $8 \pm 4\%$ in the Amazon Basin which was greater than annual natural variability of 5% (Spracklen and Garcia-Carrera, 2015). Southern and Central Amazonia regions can be considered a hotspot for changes in recycled precipitation. This is due to water limitations on ET (Biudes et al., 2015; Vourlitis et al., 2015), high precipitation recycling ratios, and the extended dry season, especially in the South-South-eastern region. Declines in precipitation can be accompanied by reductions in vegetation greenness, which further impacts the availability of green water resources and the local water balance (Hilker et al., 2014). Reduced precipitation diminishes the amount of green water available for terrestrial ecosystems with a possible impact on net primary production. Results of a five-year rainfall exclusion experiment (near CUE, Santarem, Figure 1) showed an increase in tree mortality of 5.7 % y$^{-1}$ compared to 2.5 % y$^{-1}$ for the control plot, along with a decrease in aboveground live biomass by 25% and an increasing difference in wood production between experimental and control plots by up to 58 % (Nepstad et al., 2002; Brando et al., 2008).

## 4 Including cropland irrigation in future modelling

Studies to date have considered land-atmosphere coupling in relation to agricultural expansion that is exclusively reliant on green water. However, on-farm water management can supply further water vapour to local precipitationsheds through irrigation, which also needs to be considered in future modelling work. Such consideration calls for further protection of natural ecosystems, especially in Southern and South-eastern Amazonia following climate predictions and future reductions in local regional precipitation (Coe et al., 2013). Simulations of Amazonia's possible conditions in 2050 including climate, deforestation and atmospheric feedbacks show an overall decline in aboveground biomass and agricultural yields (pasture and soybean), although the interaction of these effects are unclear at high resolution in Southern Amazonia (Oliveira et al., 2013): effects from changes to climate (+2.3°C) and atmospheric $CO_2$ concentrations (590 ppm) shortened the development cycle of soybean. This fertilization effect, however, can greatly vary based on precipitation: reduced rainfall in Southern Mato Grosso's Cerrado region likely affected soybean yields negatively (Oliveira et al., 2013). Results from such models should be confirmed with higher resolution measurements, as well as the consideration of irrigation as a viable future practice to maintain higher agricultural yields while ensuring continued water vapour supply to the atmosphere.

To illustrate this effect, we provide an estimate of irrigation vapour flows that would result from blue water consumptive use of cropland and pasture in South-eastern Amazonia. By modelling crop water requirements combining Food and Agriculture Organization guidelines (Allen et al., 1998) with meteorological data from stations located across a North to South gradient in Mato Grosso (*sensu* Lathuillière et al., 2012), we calculate these additional water vapour flows assuming crop water





requirements are fully met. We include an estimate for changes in pasture ET that would result from irrigation, assuming that 200 mm y$^{-1}$ of irrigation is supplied to the 20–22 Mha of existing pasture in Mato Grosso alone (Figure 2). These additional vapour flows thus represent an approximation of the amount of blue water redirected to ET that would be required under an ideal irrigation scenario for cropland and pasture.

The irrigation required for all cropland was estimated at 15–28 km$^3$ y$^{-1}$ during the 2000–2009 period, with higher amounts corresponding to drought periods and lower amounts related to wetter years. An additional average of 51 km$^3$ y$^{-1}$ of water vapour could have been generated between 2001 and 2009 if all cropland and pasture had been irrigated to meet water requirements. This additional blue water consumption is equivalent to about 40% of the estimated 125 km$^3$ y$^{-1}$ loss in water vapour contributions from the forest cover reduction that occurred to create cropland and pasture (Lathuillière et al., 2012).

Impacts of expanded irrigation on the local climate and precipitationsheds need to be addressed in land-atmosphere models. Additional water vapour resulting from a blue to green water transfer via irrigation under non-limiting conditions indicate that cropland alone would transfer an amount of water to the atmosphere each year equivalent to the maximum volume stored by the Itaipu dam (29 km$^3$, (Itaipu Binacional, 2015), currently the largest reservoir in Brazil. Considering local water scarcity, such consumption would have occurred at the expense of aquatic ecosystems and groundwater dependent terrestrial

ecosystems. Given the importance of the groundwater buffer for ecosystem resilience in South-eastern Amazonia (Pokhrel et al., 2014), it is also necessary to address how such blue water consumption for irrigation might impact surface and groundwater stocks and further limit ET processes.

In order to more fully consider the trade-offs between green and blue water resulting from land use change, climate change and alterations in water management including irrigation, we consider the interacting effects of agricultural expansion into

natural ecosystems in South-eastern Amazonia that contribute to decreasing ET and precipitation recycling in the broader region. We propose five possible options for land and water management for future agricultural production in the region considering the current objectives to increase agricultural production (MAPA, 2013). These options include the expansion of rain-fed agriculture into natural ecosystems (option A), the expansion of rain-fed agriculture into current pastureland (option B), improved soil water management to reduce evaporation and increase transpiration, or vapour shift (option C), rainwater

harvesting (option D), and the expansion of irrigation in current production (option E) (Table 3).

From all the water management options proposed, expansion of rain-fed agriculture into natural ecosystems (options A), rainwater harvesting (option D) and irrigation expansion (option E) will result in trade-offs with blue water users downstream. Expansion of rain-fed agriculture into current pasture (option B) and vapour shift (option C) will not incur such trade-offs while still possibly maintain precipitation recycling. Expansion of rain-fed agriculture into natural ecosystems (option A) is

the only presented option that would further call for deforestation with consequences on the water cycle, but also biodiversity and CO$_2$ emissions. Expansion into current pasture (option B) still promotes green water consumption through expansion by colonizing current pastureland. A combination of horizontal expansion of cropland into pasture and irrigation (options B and E) would considerably increase the amount of surface and groundwater required for agriculture. A doubling of 2013 soybean production (from 23 Mtons according to IBGE (2015)) would require roughly 7.5 Mha of pastureland to be converted to



soybean with an additional 10.5 km$^3$ y$^{-1}$ of blue water required for irrigation. Options C, D, and E represent intensification options with and without blue water consumption, respectively by vapour shift, rainwater harvesting and expansion of irrigated land. These options might be more desirable given Brazil's objectives to reduce deforestation by 80% in the Amazon by 2020 and 40% in the Cerrado compared to a 1996–2005 baseline (Galford et al., 2013). To some extent, rainwater harvesting has

already taken place in the case of cattle ranching on pastureland when small, often rain-fed farm dams, are used to supply cattle with drinking water. In 2007, about 10,000 such dams were accounted for in the Upper Xingu Basin of Mato Grosso (Macedo et al., 2013). Considering an average farm dam size of 0.25 km$^2$ and a volume to area relationship similar to the state of Goiás described in Rodrigues et al. (2012), we estimate a total small farm dam volume of 6 km$^3$, of which 2.4 km$^3$ may be directly evaporated according to small dam evaporation estimates (Baillie, 2008). Such additional water vapour supply to the

atmosphere should also be accounted for in future models. The case of irrigation expansion (option E) has the potential to promote agricultural intensification with marginal improvements to precipitation recycling which still needs to be proven in future research. While this option does prevent further deforestation, its impact on surface and groundwater resources will have to be assessed to identify a win-win scenario of increased agricultural production, precipitation recycling without degradation of aquatic ecosystems.

**5 Conclusions**

This review provides a detailed assessment of precipitation partitioning of Amazonia and South-eastern Amazonia's water resources into green and blue water and considers important questions about the future of land and water management in the basin. The current state of knowledge on precipitation, ET and discharge in the basin as well as the possible effects of land use change on the hydrological cycle create additional unknowns in a region that is expected to transform from direct human

involvement in land use management or indirectly from global climate change. The field of global ecohydrology can play an important role in understanding how we can limit the impact of future economic development and land management on the hydrological cycle.

South-eastern Amazonia was identified as an important region for future land and water use planning based on the following: (1) its role in future agricultural expansion for pasture and soybean, (2) the region's reliance on water for ensuring ecological

and agro-ecological functions, (3) the importance of precipitation recycling and its emerging connection to land use change which might affect other Amazon sub-basins, and (4) the potential of irrigation expansion to maintain production and prevent further encroachment of agriculture into natural ecosystems. The best land and water strategy will be one that ensures connectivity within the hydrological cycle, minimizes up- and downstream blue water trade-offs from agricultural production and maintains precipitation recycling in the region to prevent future degradation of natural ecosystems. Policy options should

consider a combination of expansion of soybean into pastureland as well as increases in rain-fed agricultural yields either through an improvement in productive green water use or proper rainfall harvesting. Meat production in Brazil could still be increased without further deforestation, mainly by increasing beef yields (Strassburg et al., 2014), while rainwater harvesting



could be used to ensure supplemental irrigation in the second crop (e.g. maize or cotton) typically harvested at the onset of the dry season. This strategy has the benefit to conserve biodiversity and prevent further greenhouse gas emissions from deforestation.

Finally, such strategies have to be incorporated into greater objectives such as enforcement of the Federal Forest Code, soybean and beef supply chain interventions and land tenure issues in Brazil (Lapola et al., 2013; Nepstad et al., 2014). Global ecohydrology has a role to play in complementing these strategies to secure future water resource needs of the rapidly development Amazon region.

**Acknowledgments**

This review represents a contribution to "Integrating land use planning and water governance in Amazonia: Towards improving freshwater security in the agricultural frontier of Mato Grosso" a project supported by the Belmont Forum and the G8 Research Councils Freshwater Security Grant G8PJ-437376-2012 through the Natural Sciences and Engineering Research Council (NSERC) to MSJ, and the National Science Foundation to MTC. Additional support was provided by the Vanier Graduate Scholarship through NSERC to MJL (#201411DVC-347484-257696). Special thanks are owed to George Vourlitis for insightful discussions.

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



**Table 1 Representation of water resource terminology in the green and blue water ecohydrological approach (Falkenmark and Rockström, 2004, 2006).**

| Water | Stock | Flow | Consumptive pathway |
|---|---|---|---|
| Precipitation | Atmosphere | Liquid, Solid | Source |
| Blue | Runoff, rivers, reservoirs, wetlands, lakes, snowpack, aquifers | Liquid | Household or industrial uses, drinking water, product integration |
| Blue | Surface or groundwater | Vapor | Evapotranspiration from irrigation |
| Green, Productive green | Soil moisture | Vapor | Plant transpiration |
| Green, Unproductive green | Soil moisture, intercepted rainfall | Vapor | Evaporation (soil, surface, snow) |



**Table 2: Evapotranspiration of biomes, cropland (green and blue water) and pasture derived from remote sensing as described in the references**

| Humid tropical forest | Deciduous broadleaf forest | Savannah | Cropland | Grassland | References |
|---|---|---|---|---|---|
| | | | *mm y⁻¹* | | |
| 1245 | 792 | 882–1267 | 1438 | 599 | (Rockström et al., 1999) |
| 800 | | 280–1200 | | | (Falkenmark and Rockström, 2004) |
| 1310–1675 | 635 ± 200 | 676 ± 183 | 507 ± 157 | 311 ± 193 | (Bruijnzeel, 1990; Zhang et al., 2010) |
| 1182 | 200–900 | 806 | 542 | 462 | (Miralles et al., 2011 ; Matyas and Sun, 2014) |




**Table 3: Green and blue water use options to increase agricultural production in South-eastern Amazonia with possible trade-offs in water resources and regional precipitation recycling.**

| Option | Strategy | Effects on agricultural production | Effect on water resources in the region | Possible effects on precipitation recycling |
|---|---|---|---|---|
| A. Increase green water use for agriculture | Expansion of rain-fed agriculture and pastureland into natural ecosystems | Increase production by area | Reduced overall ET; trade-offs expected with blue water users downstream | Reduced |
| B. Increase green water use for agriculture | Expansion of rain-fed agriculture into pastureland | Increase production by area | Marginal change in overall ET; blue water downstream unchanged | Maintained |
| C. Increase green water use for agriculture | Vapor shift from evaporation to transpiration to improve productive green water use | Increase production by yield | Improves productive green water use and yields by postponing possible future irrigation (blue water savings); blue water downstream unchanged | Maintained |
| D. Increase green water use for agriculture | Rainwater harvesting used off season | Increase production by yield | Improves green water consumptive use in the same location as precipitation; trade-offs expected with blue water users downstream | Maintained |
| E. Increase in blue water use | Blue water used to irrigate agriculture and prevent further expansion into natural ecosystems | Increase production by yield | Possible impacts on aquatic ecosystems from the consumption of surface and groundwater; trade-offs expected with blue water users downstream | Increased |





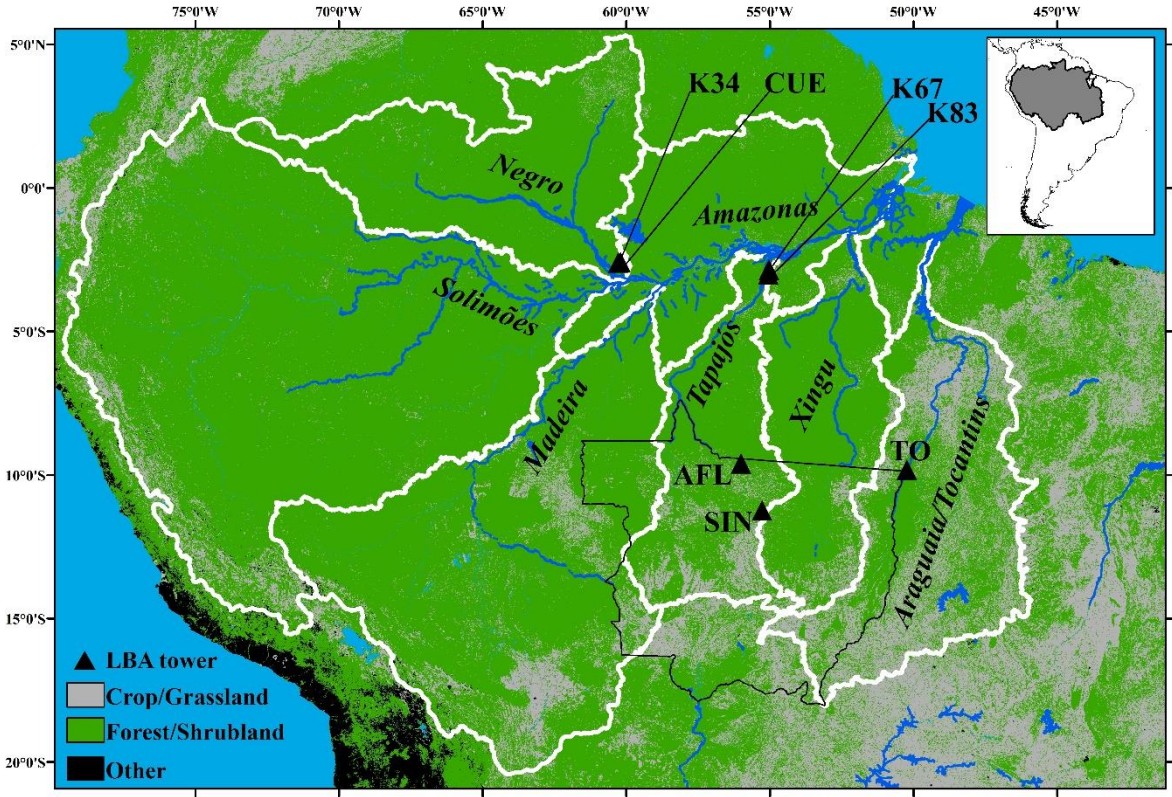

**Figure 1: The Amazon Basin of South America with its main river basins (ANA, 2015), eddy covariance tower network from the Large-Scale Biosphere-Atmosphere Project in Amazonia (Table S3) (Keller et al., 2004) and aggregated land uses as classified by the ESA GlobCover 2009 Project (ESA, 2010; ©ESA 2010 and UCLouvain) and the political divide of the Brazilian state of Mato Grosso in South-eastern Amazonia.**




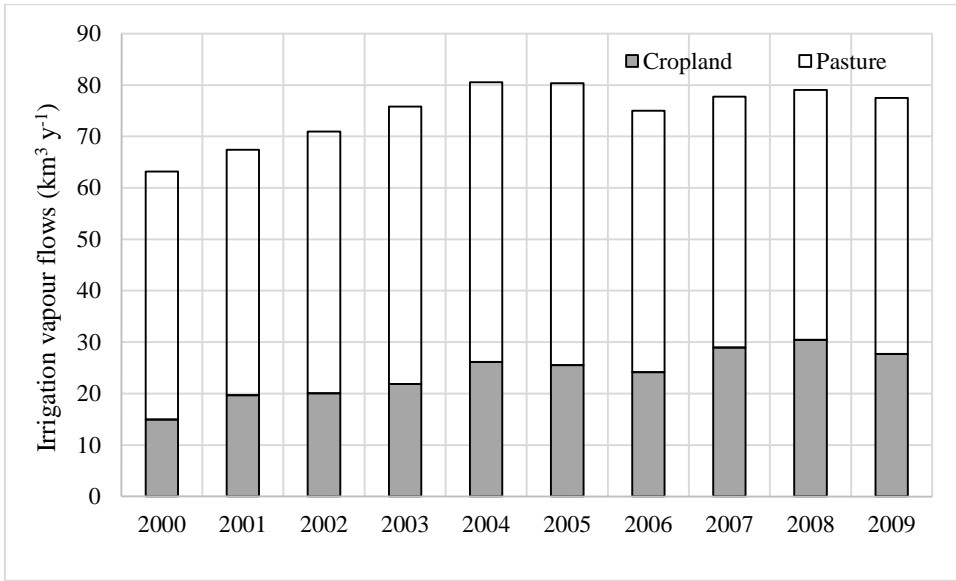

**Figure 2: Annual irrigation requirements for cropland (as the sum of soybean, maize, cotton and sugar cane) and pasture in South-eastern Amazonia (Mato Grosso) for the 2000–2009 period.**