# Peer review of "A review of green and blue water resources and their trade-offs for future agricultural production in the Amazon Basin: What could irrigated agriculture mean for Amazonia?"

_Hydrology and Earth System Sciences, 2016_

## Referee Comment (RC1) · Anonymous Referee #1 · 15 Mar 2016

General comment:

This manuscript provides an overview on the water balance, and human effects on it, in the Amazon basin. Besides a rough analysis of possible future irrigation water requirements it does not present original research but appears to be designed as a review paper. That might be fine if HESS accepts such paper formats, and I do not have any major concerns about the presented summary of previous studies. However, even if this HESSD paper considered a review paper, I have several suggestions for improvement and focus of the final HESS, as follows.

Major comments:

The discussion of green vs. blue water (entire section 2) is much too lengthy; basically it provides a summary of ideas by Rockström, Falkenmark and colleagues which can be read elsewhere: I suggest to cut major parts of this section, only briefly summarize it and focus it more directly on the study area (Amazon) while if possible leaving out discussion of findings from other areas (like the Zeng et al. analysis etc.). There are only two figures in the main document but many more in the Supplement; why not include e.g. Table S3 and S4 in the main text? They are at least as informative as Figs. 1 and 2, especially since they include numbers on the Amazon which is your study area for this paper. I also think that some of the Amazon-based discussion in the SI belongs to the main text, while the discussions of global ET etc. could be moved to the SI or removed as a whole. Besides I miss a figure with any quantitative information on the (green-blue) water flows and the impacts of e.g. land use change, which would be very helpful as on overview and to support the four main conclusions from section 5. Otherwise it is difficult for readers to extract the main observations and findings from sections 3 and 4 and assign them to specific regions within the Amazon basin. What is still lacking is a good and illustrative overview of the main findings / concerns.

Technical/minor comments:

Title: The first part on the possible role of irrigation is a bit misleading, as this discussion is only a smaller part of this paper, and thus it could be removed (the title is too long anyway). In general (in the Abstract and the Intro) the scope of the paper could be made even clearer, i.e. pointing out that it is a review of previous studies. In line with my above comment on being clearer about the scope and structure of the paper: At the end of the Intro it should be stated what irrigation scenario is analysed here. At the end of section 2 it should be clarified how you "explore these implications. . .". Tables 1 and 2 are not very informative nor do they say anything about the Amazon region – remove?

---

## Referee Comment (RC2) · Anonymous Referee #2 · 22 Mar 2016

General comments:

This paper provides a comprehensive review of Amazonian, and specifically south and south-eastern Amazonian, water resources within the framework of green and blue water accounting. This guiding framework unites a large and disparate body of work on land use, ET, precipitation, and river runoff, and connects that work to forest and agricultural water management options and future research. With minor changes, primarily to improve clarity, this review will be useful to those working at the intersection of land use, climate, and hydrology in Brazil.

Specific comments:

The title's focus on irrigation is not entirely representative of the focus of the review, which is broader. I'd recommend cutting "What could irrigated agriculture mean for Amazonia?" from the title.

In general, the introduction provides adequate motivation, but does not provide a clear roadmap and summary of what this review accomplishes (see technical corrections).

Use of the term 'ecohydrology' to describe the approach and/or framework of the paper may be confusing to some readers. My understanding is that the body of work the authors reference may be more commonly known as the 'water footprinting' framework (or just blue and green water accounting). My understanding is that the field of eco-hydrology is more specifically focused on climate-soil-vegetation dynamics and related theory, the literature for which is not referenced in this paper. See: (1) Rodriguez-Iturbe, I. "Ecohydrology: A Hydrologic Perspective of Climate-Soil-Vegetation Dynamies." Water Resources Research 36, no. 1 (2000): 3–9. doi:10.1029/1999WR900210; (2) Rodríguez-Iturbe, Ignacio, and Amilcare Porporato. Ecohydrology of Water-Controlled Ecosystems: Soil Moisture and Plant Dynamics. Cambridge: Cambridge University Press, 2005.

The back-of-the-envelope calculations nicely motivate the call for future research on irrigation, but their presentation is a bit simplistic compared to the detail of the preceding literature review. Two brief discussion items could help address this issue - those are described below.

First, there is debate about the quantification and efficacy of improvements to productive green water use (option C - what is also commonly referred to as agricultural water use efficiency and productivity). Debates on this matter tend to arise in local contexts, but are relevant to any policy proposal that seeks to address water management problems with irrigation. See for example a California, USA case wherein the benefits of irrigation efficiency improvements are debated on theoretical grounds: (1)

Frederiksen, Harald Dixen, and Richard Glen Allen. "A Common Basis for Analysis, Evaluation and Comparison of Offstream Water Uses." Water International 36, no. 3 (2011): 266–82. doi:10.1080/02508060.2011.580449; and response (2) Gleick, Peter H., Juliet Christian-Smith, and Heather Cooley. "Water-Use Efficiency and Productivity: Rethinking the Basin Approach." Water International 36, no. 7 (2011): 784–98. doi:10.1080/02508060.2011.631873.

Second, there is no mention of water quality or 'grey water' trade-offs implicit in policy options that exploit trade-offs between green and blue water to address water quantity challenges. All but one policy option (D) are likely to result in some impact to water quality (pollutant loads in soils and streams). This is worth mention, as green, blue, and grey water trade-offs should be included in any analysis that weighs the costs and benefits (to people and the environment) of different policy options.

Technical corrections:

Page 2, lines 11-12 and 30-33: I don't think these sentences summarize what this review does. I don't think the authors "evaluate this framework" in the contexts described (line 11-12), or "review what the green and blue water ecohydrological approach can bring...", but instead use the concept of blue and green water to frame a literature review and discussion of proposed water management and policy options, and to guide future research.

Page 3, line 30: awkwardly phrased and/or typo at "a blue water redirect"

Page 5, line 1: the Rˆ2 given for relationship between latent heat and VPD and NDVI do not show "strong correlation" - or if they do, the way this sentence and surrounding sentences are written is confusing. The next sentence (lines 1-3) makes sense for the first study referenced (Zeng et al., 2012), but not necessarily for the second (Fisher et al., 2009). I don't entirely understand the point the authors are trying to make other than that there is a mechanistic relationship between surface energy budgets, water availability, and vegetation which is shown in empirical studies.

Page 5, line 7: Perhaps consider inserting something like "consumption" or "outflow" after "Declines in green water", or replacing "green water" with ET, for clarity.

Page 5, line 10-11: This sentence is a little unclear - perhaps (1) insert "at regional scales" after "precipitation recycling" and (2) replace "affect" with "decrease" to make it clear that the opposite of the previously described dynamic can occur because of feedbacks at multiple scales.

Page 5, line 18: The wording "a global compensation of ET with land cover and uses" is confusing. Perhaps consider instead something line "only a small net loss due to land use and cover change"

Page 5, line 26 - Page 6, line 29. The first part of this section appears to be a summary review of the global literature, which then transitions into a discussion of blue and green water trade-offs in south-eastern Amazonia. The relationship between the points made in the first part (global summary) and the latter part (SE Amazon), and more specifically the transition, is not entirely clear. On page 6, line 29 a contrast is made: "... different green and blue water realities than those described above" and it's not clear if the "above" are the global findings in general, or the countries referenced on page 6, lines 21-22 and 26. If the point is that the global findings are not especially relevant to the Amazon region, perhaps this section could be shortened, and only findings that are relevant (either directly, or in contrast) to Amazonia could be discussed since the global literature on green and blue water accounting has been summarized extensively elsewhere. Lastly, if the discussion of global findings are retained - on page 6, line 26-27, relevant (and more recent) literature includes: (1) Suweis, Samir, Joel A. Carr, Amos Maritan, Andrea Rinaldo, and Paolo D'Odorico. "Resilience and Reactivity of Global Food Security." Proceedings of the National Academy of Sciences 112, no. 22 (June 2, 2015): 6902–7. doi:10.1073/pnas.1507366112; (2) Suweis, Samir, Andrea Rinaldo, Amos Maritan, and Paolo D'Odorico. "Water-Controlled Wealth of Nations." Proceedings of the National Academy of Sciences 110, no. 11 (March 12, 2013): 4230–33. doi:10.1073/pnas.1222452110.

Page 7, line 19: Given the Rˆ2 value, perhaps delete "highly".

Page 8, line 28: delete "the" before "1988-2014", or add "period" after.

Page 12, line 13: either a missing or extra parenthesis.

---

## Author Comment (AC1) · 24 Mar 2016

We thank you for your comments on how to improve our paper. This paper is indeed meant as a Review with our main objective to think about what may come for Amazonia's agriculture water use by bringing together two main research themes. From our experience, the theme of "green/blue water" and "Amazonia water resources" haven't really coexisted before in a paper and therefore we expect that the academic literature of one theme to be somewhat new for experts in the other theme.

Major comments:

-"I suggest cutting out major parts of [section 2], only briefly summarize it and focus it more on directly on the study area"

We have structured the paper to provide equal weight to both themes with part 2 (green/blue water) and 3 (Amazonia's water resources) being roughly 2400 words each. As part 2 focuses more on the green/blue ecohydrological perspective, we felt compelled to provide a proper, and up to date review, of what this perspective is and means for water resources prior to applying the perspective to the Amazonia context. As such, we also felt it was important to briefly update the information on global evapotranspiration models and their findings (e.g. Zeng et al., 2012) as a precursor to highlighting differences between global and regional contexts. That being said, we do see some possible improvements or combination of sections 2.2 and 2.3 with some more summarized information that can shorten section 2 overall and give more space to the description of the study region, without compromising information.

-"Why not include tables S3 and S4 in the main text?"

Thank you for this comment and perspective on providing more quantitative information in the Review. Table S3 could be brought into the main text to accompany Figure 1, while moving Table 2 in the Supplemental Material during our attempt to reduce section 2.

-"I also think that some of the Amazon-based discussion in the SI belongs to the main text, while discussion on global ET could be moved to the SI"

Originally much of the discussion on Amazonia in the SI was in the main text and we found it to be distracting for our main purpose of the paper to review the region's water resources in a new light.

-"I miss a figure with any quantitative information on the (green-blue) water flows and the impacts of e.g. land use change"

Thank you for your suggestion, we think this is an excellent idea and we'll consider it in
the revised manuscript.

Technical/minor comments

-"Title: The first part on the possible role of irrigation is a bit misleading, as this discussion is only a smaller part of this paper, and thus could be removed" and "the scope of the paper could be made even clearer"

As described above, our review aims to bring together two research themes in order to shed new light on the future of water resources in Amazonia, particularly irrigation which has not been widely developed in the region. We wanted our title to clearly identify the main question we have on our minds before providing an avenue (the green/blue water perspective) to assess the future of water resources in Amazonia.

---

## Author Comment (AC2) · 24 Mar 2016

Specific comments:

-"I'd recommend cutting "What could irrigated agriculture mean for Amazonia?" from the title

This is now the second comment we've received regarding the title. As discussed with anonymous reviewer 1 (AR1), we were hoping to bring forward our main question about additional water vapour flows that would be generated by irrigation within the more general aspect of agricultural water management. We are now considering moving the

position of the question to follow the outline of the paper and suggest this new title: "A review of green and blue water resources and their trade-offs for future agricultural production in the Amazon Basin: What could irrigated agriculture mean for Amazonia?"

-"My understanding is that the body of work that the authors reference may be more commonly known as the 'water footprinting' framework (or just blue and green water accounting)"

We view the separation of green/blue water as essentially an ecohydrological perspective because: (1) it provides a new focus on the role of soil moisture in what Rodrigues-Iturbe (2000) described as a "keystone of numerous fundamental questions which may be instrumental in the quantitative linkage between hydrologic dynamics and ecological patterns and processes" (Water Resources Research 36(1), 3–9, doi:10.1029/1999WR900210) and includes the recycling of regional precipitation as a key ecosystem service (Ellison, D. (2012) On the forest cover-water yield debate: from demand- to supply-side thinking, Global Change Biology, 18, 806‒820, doi: 10.1111/j.1365-2486.2011.02589.x.); (2) it provides a much needed framework to link water and carbon cycles in the context regional and global challenges (Dolman, A.J. et al. (2014) Fifty years since Monteith's seminal paper: the emergence of global ecohydrology, Ecohydrology 7, 897-902, doi: 10.1002/eco.1505).

It is true that the terminology of green/blue water appears frequently in the 'water footprint framework' which focuses exclusively on human appropriation of water resources and does not necessarily include natural ecosystems in the same way the green/blue water literature cited in our manuscript has addressed it. A water footprint assessment is a logical consequence of what we are describing in this manuscript and is currently in preparation for the region. At this time, we feel that adding another research "theme" and terminology to the paper could bring about more confusion. This would be particularly true for an audience more familiar with Amazonia's water resources which does not necessarily use terms such as as green or blue water.

-"The back-of-the-envelope calculation nicely motivate the call for future research on irrigation (. . .) First, there is a debate about the quantification and efficacy of improvements to productive green water use (. . .); Second, there is no mention of water quality trade-offs.

We appreciate these comments based on experience in different regional contexts from which our manuscript could benefit some nuances in the two points you bring up. There are still many unknowns in water use for agriculture in South-Southeastern Amazonia that many details will be missing when looking at the different options in Table 3. So far, most (if not all) ET estimates for agricultural water use (and pasture) in Mato Grosso are based on models rather than measurements published in peer-reviewed journals. This is not the case for natural ecosystems which have been studied (and continue to be) as shown in Table S3. With this discrepancy in mind, and given that agriculture is almost entirely rain-fed, it is difficult today to understand exactly how much more yield improvement could be achieved through green water management, especially for soybean, maize, cotton, sugar cane, and rice. Our experience tells us that yields will likely increase due to other inputs (e.g. fertilizer or genetics), although we are starting to see shifts in water management that could be quite significant.

Second, you are absolutely right to mention water quality aspects which have not been addressed in this paper, again for lack of information on this point. We'd expect the addition of lime, fertilizer and pesticides to soybean, pasture, maize and cotton fields to have impacts on nearby water bodies. The case of eutrophication comes to mind with the average state wide application of 0-5 kg N ha$^{-1}$ and 28-34 kg P ha$^{-1}$ of fertilizer in the case of soybean (Lathuillière, M.J. et al. (2014) Environmental footprints show China and Europe's evolving resource appropriation for soybean production in Mato Grosso, Brazil, Environmental Research Letters 9(7), 074001, doi: :10.1088/1748-9326/9/7/074001) but so far, field studies have not seen this leading to water quality impacts in Mato Grosso (e.g. Riskin, S. et al. (2013) The fate of phosphorous fertilizer in Amazon soya bean fields, Phil. Trans. R. Soc.

B 368, 20120154, doi: 10.1098/rstb.2012.0154). Possible irrigation, especially in the dry season in Southeastern Amazonia, would likely increase any pollution load currently assimilated by soils, perhaps with even greater consequences on aquatic ecosystems due to lower streamflow during the dry season. So far, measurements made by the Environmental Secretariat of Mato Grosso (SEMA) designate water quality of most rivers upstream of the Amazon river between 2012 and 2014 as "GOOD" (or a mark of 4/5) or "REGULAR" (3/5) with monthly variations (SEMA, Relatório de Monitoramento de Qualidade da Água – Região Hidrográfico Amazônica – 2012-2014, http://www.sema.mt.gov.br/index.php?option=com_docman&Itemid=82).

In short, we will add both of these helpful points to the discussion of Table 3 in the revised manuscript in order to nuance the proposed options.

Technical corrections:

Page 2, lines 11-12 and 30-33: Thank you, we will review this sentence entirely.

Page 3, line 30: We will review this sentence.

Page 5, line 1: Here, we illustrate the mechanisms controlling evapotranspiration (ET) which can be atmospheric or biological in nature (and both exist across the Amazon region). You are right in pointing out that there is no strong correlation when looking at NDVI or VPD; this comment was directed to net radiation only when looking at Zeng et al. (2012), so we will have to rephrase slightly here. Also, we believe that the confusion in this paragraph comes from the different scopes of Zeng et al. (2012) (global) and Fisher et al. (2009) (tropical regions) which show different mechanisms. We will have to clearly separate these two papers to reduce confusion in the results that they are showing.

Page 5, Line 7: Thank you, we will consider this in the revised version of the manuscript. We have

been very careful in the paper to refer to green water consumption as ET and avoid

the term "green water flow" which we find does not represent the physically measured process.

Page 5, Line 10-11: We agree with these suggestions.

Page 5, Line 18: We agree with this suggestion.

Page 6, Line 26 to Page 6, Line 29: We seek to highlight an important parallel for the application of the green/blue water perspective. The perspective has often been useful in the Sub-Sahara African context with the proposal to upgrade rain-fed agriculture to improve food security. We argue here, that the perspective can also be useful in regions of similar climate to Sub-Saharan Africa but considering different future agricultural production options. The references you have provided merit some attention before refocusing this paragraph to smooth the transition we want to make between global analysis of water use for agriculture, the Sub-Saharan Africa context and a "new" context in Southeastern Amazonia.

Page 7, Line 19: We agree with this suggestion.

Page 8, Line 28: We agree with this suggestion.

Page 12, Line 13: Thank you.

---

## Author Response (AR1)

May 6, 2016

Dr. Sally Thompson
Associate Editor, *Hydrological and Earth Systems Sciences*

Dear Dr. Thompson,

Thank you coordinating the review of our manuscript hess-2016-71 entitled "What could irrigated agriculture mean for Amazonia? A review of green and blue water resources and their trade-offs for future agricultural production in the Amazon Basin" submitted to HESS. We have now made changes to our manuscript following suggestions and comments from the reviewers.

We have changed the title to "A review of green and blue water resources and their trade-offs for future agricultural production in the Amazon Basin: What could irrigated agriculture mean for Amazonia?" to reflect the organization of our paper. The article is destined as a Review with two main objectives: 1) to bring together the ecohydrological perspective proposed by the green/blue water research theme with water resources in Amazonia, and 2) to use this description of water resources to answer an important question about the future of water use in Amazonia following continued growth in agricultural production.

Following comments about reorganizing section 2, we have shortened the section by 800 words as a means to reduce the discussion on the green/blue water perspective thereby adding more focus to Amazonia. Section 2 now carries 3 sub-sections rather than 4 as we combined what was previously "2.2. Green water resources, vapour flows and precipitation" with "2.3. Precipitation partitioning on land and green/blue water trade-offs" into a new section "2.2. Precipitation recycling and green/blue water trade-offs". That section was made more concise by removing text from the original submission based on detailed discussion in other review papers as per recommendation from the Reviewers. Finally, we have also removed what was previously the final paragraph of section 2.4. introducing our region of interest, and have placed it at the beginning of section 3.1. in a new sub-section entitled "3.1. Brazil's agricultural frontier of Southern Amazonia".

We have also brought information out of the Supplemental Material to highlight quantitative information available for Amazonia, as per Reviewer comments. Table 2 now shows the change in precipitation partitioning across Amazonia while Table 3 summarizes precipitation recycling ratios in the region. We have also added a new Figure 3 to graphically illustrate the land management options proposed in Table 3 with some quantitative examples of evapotranspiration and runoff based on local water balance results.

Please find below more detailed comments regarding major and minors comments from both Reviewers 1 and 2 (highlights in blue). All changes made in the manuscript are highlighted in yellow with additional notes appearing as comments in the Word file.

We look forward to hearing from you,

Best Regards,

Michael Lathuillière.
mlathuilliere@alumni.ubc.ca

**Responses to Reviewer comments**

*Referee 1 – Major comments*

-"I suggest cutting out major parts of [section 2], only briefly summarize it and focus it more on directly on the study area"(…)

    -As discussed above we have made some changes to part 2.2 and 2.3 in the text while still maintaining important information about the green/blue water framework. As suggested by the Reviewer, we have moved what were previously Tables S3 and S4 to the main text as well as a new Figure 3 with quantitative information summarizing the land and water management options presented in Table 3.

*Reviewer 1 - Minor comments*

-"Title: The first part on the possible role of irrigation is a bit misleading, as this discussion is only a smaller part of this paper, and thus could be removed" and "the scope of the paper could be made even clearer"

    -We have placed the questions of the title ("What could irrigated agriculture mean for Amazonia?") in second place in order to follow the outline of the paper as described above.

*Reviewer 2 – Major comments*

-In general, the introduction provides adequate motivation, but does not provide a clear roadmap and summary of what this review accomplishes (see technical corrections).

    We have now included an adequate roadmap at the end of the introduction.

-Use of the term "ecohydrology" vs. "water footprint"

    -We retained the description of water resources by using the green/blue water "ecohydrological perspective" rather than the "water footprint approach" as we believe the ecohydrological approach can better address the use of water by terrestrial ecosystems as opposed to the "water footprint" approach which focusses exclusively

on human appropriation of water. We are currently working on a Water Footprint
Assessment for the region as a logical sequence of what we describe in this manuscript

-"The back-of-the-envelope calculation nicely motivate the call for future research on
irrigation (…) First, there is a debate about the quantification and efficacy of improvements
to productive green water use (…); Second, there is no mention of water quality trade-offs.
(…)

-We agree with the author that some nuance was warranted here and have made
additional comments to address the issues of agricultural water use efficiency (option
C in Table 3) on p.12 line 25, as well as a few additional sentences on water quality
aspects at the end of section 4.

*Reviewer 2 - Technical corrections*

-Page 2, lines 11-12 and 30-33: I don't think these sentences summarize what this review
does. I don't think the authors "evaluate this framework" in the contexts described (line 11-
12), or "review what the green and blue water ecohydrological approach can bring...", but
instead use the concept of blue and green water to frame a literature review and discussion
of proposed water management and policy options, and to guide future research.

-We have changed the wording of this sentence to highlight our proposal to apply the
green/blue water framework described in the introduction rather than "evaluating"
the framework as previously outlined.

-Page 3, line 30: awkwardly phrased and/or typo at "a blue water redirect"

-We have changed this phrase to "resulting from blue water consumption"

-Page 5, line 1: the $R^2$ given for relationship between latent heat and VPD and NDVI
do not show "strong correlation" - or if they do, the way this sentence and surrounding
sentences are written is confusing(…)

-We have reviewed the sentence to eliminate confusion.

-Page 5, line 7: Perhaps consider inserting something like "consumption" or "outflow" after "Declines in green water", or replacing "green water" with ET, for clarity.

-This change has been made.

-Page 5, line 10-11: This sentence is a little unclear - perhaps (1) insert "at regional scales" after "precipitation recycling" and (2) replace "affect" with "decrease" to make it clear that the opposite of the previously described dynamic can occur because of feedbacks at multiple scales.

-This change has been made.

[revised manuscript text omitted]

---

## Author Response (AR2)

May 21, 2016

Dr. Sally Thompson
Associate Editor, *Hydrological and Earth Systems Sciences*

Dear Dr. Thompson,

Thank you for your feedback regarding corrections to be made to our paper entitled "A review of green and blue water resources and their trade-offs for future agricultural production in the Amazon Basin: What could irrigated agriculture mean for Amazonia?" (hess-2016-71).

I have now read the paper thoroughly to ensure that all acronyms were defined, specifically those on lines 15 (SEMA) and 19 (CONAMA) of page 13 when referring to the State and Federal environmental councils in Brazil. I have also made very minor typo corrections in the Supplemental Material and the references.

We look forward to the final published version of the manuscript,

Best Regards,

Michael Lathuillière.